# Is Blood Eosinophil Count a Biomarker for Chronic Obstructive Pulmonary Disease in a Real-World Clinical Setting? Predictive Property and Longitudinal Stability in Japanese Patients

**DOI:** 10.3390/diagnostics11030404

**Published:** 2021-02-27

**Authors:** Koichi Nishimura, Masaaki Kusunose, Ryo Sanda, Mio Mori, Ayumi Shibayama, Kazuhito Nakayasu

**Affiliations:** 1Department of Respiratory Medicine, National Center for Geriatrics and Gerontology, Obu 474-8511, Japan; kusunose@ncgg.go.jp (M.K.); ryo-sand@ncgg.go.jp (R.S.); mio-mori@ncgg.go.jp (M.M.); 2Department of Nursing, National Center for Geriatrics and Gerontology, Obu 474-8511, Japan; ayuminarita3@ncgg.go.jp; 3Data Research Section, Kondo Photo Process Co., Ltd., Osaka 543-0011, Japan; nakayasu@mydo-kond.co.jp

**Keywords:** inhaled corticosteroids (ICS), chronic obstructive pulmonary disease (COPD), blood eosinophil count (BEC), eosinophil, the Global Initiative for Chronic Obstructive Lung Disease (GOLD), acute exacerbation of COPD (AECOPD)

## Abstract

The authors examined predictive properties and the longitudinal stability of blood eosinophil count (BEC) or three strata (<100 cells/mm^3^, 100–299 cells/mm^3^ and ≥300 cells/mm^3^) in patients with chronic obstructive pulmonary disease (COPD) for up to six and a half years as part of a hospital-based cohort study. Of the 135 patients enrolled, 21 (15.6%) were confirmed to have died during the follow-up period. Episodes of acute exacerbation of COPD (AECOPD) were identified in 74 out of 130 available patients (56.9%), and admission due to AECOPD in 35 out of 132 (26.5%). Univariate Cox proportional hazards analyses revealed that almost all the age, forced expiratory volume in 1 s (FEV_1_) and health status measures using St. George’s Respiratory Questionnaire (SGRQ) Total and COPD Assessment Test (CAT) Score were significantly related to these types of events, but the relationship between age and AECOPD did not reach statistical significance (*p* = 0.05). Neither BEC nor the three different groups stratified by BEC were significant predictors of any subsequent events. There were no significant differences in the BEC between Visits 1–3 (*p* = 0.127, Friedman test). The ICC value was 0.755 using log-transformed data, indicating excellent repeatability. In the case of assigning to strata, Fleiss’ kappa was calculated to be 0.464, indicating moderate agreement. The predictive properties of BEC may be limited in a real-world Japanese clinical setting. Attention must be paid to the fact that the longitudinal stability of the three strata is regarded as moderate.

## 1. Introduction

The question of whether inhaled corticosteroids (ICS) should be administered to patients with chronic obstructive pulmonary disease (COPD) has been debated for over three decades [1,2,3,4]. It could not have been expected that blood eosinophil count (BEC) would emerge at the heart of this debate. Some post-hoc analyses of relatively large-scale clinical trials for studying ICS-containing regimens in patients with moderate and severe COPD have reported that the BEC is significantly able to predict the response to ICS since this medication was most efficacious in the prevention of exacerbation in patients with higher baseline BEC [5,6,7]. This hypothesis was subsequently investigated in the development procedures of the single-inhaler triple therapy and the BEC was thus established as a prognostic biomarker [8,9,10]. The Global Initiative for Chronic Obstructive Lung Disease (GOLD) document has changed to reflect these new findings, especially in Group D, and currently reports that ICS-containing regimens have little or no effect at a blood eosinophil count of <100 cells/mm^3^, and that a threshold blood eosinophil count of ≥300 cells/mm^3^ or frequent exacerbation with a threshold blood eosinophil count of ≥100 cells/mm^3^ can be used to identify patients with the greatest likelihood of treatment benefit with ICS [11]. 

However, the health indicators including biomarkers should be discussed from the following three different perspectives. First, they can differentiate between people who have better health and those with worse health (a discriminative property). Second, they can measure how much the health condition changes (an evaluative property). Third, they can predict the future outcomes of patients (a predictive property). Therefore, to determine whether or not the BEC can be regarded as a biomarker in COPD, multifaceted analysis and evaluation as an outcome marker will be required.

Compared with western countries, ICS may have been less preferably prescribed for patients with COPD in Japan. In the 5th version of the Japanese guidelines published by The Japanese Respiratory Society in 2018, the description reads that ICS should be given only in patients with asthma-complicated COPD or Asthma and COPD Overlap (ACO) [12]. It is reported that the blood eosinophil data from global studies are of relevance in Japan although there was a slightly lower median eosinophil count for Japanese patients within multi-country studies [13]. One of the opposing views is that BEC may be liable to variation and considered to be unreliable as a biomarker [14,15,16,17,18,19,20,21,22,23,24]. The accuracy and diagnostic value of the BEC may be critical to the selection of appropriate ICS-containing treatments and should continue to be studied also in Japan.

The authors hypothesized that BEC could predict exacerbation or other subsequent events even in real-world clinical practice since it has been reported that possible reduction of the future acute exacerbation of COPD (AECOPD) by ICS is related to the blood eosinophil count. In addition to analysis of the absolute number of the BEC, the counts are divided into the following three groups according to GOLD 2019 thresholds: non-eosinophilic, intermediate and eosinophilic defined as BEC <100 cells/mm^3^, 100–299 cells/mm^3^ and ≥300 cells/mm^3^, respectively. We aimed to investigate how BEC, or the three strata are cross-sectionally related to other clinical measures at baseline and to examine predictive properties of the baseline values regarding mortality, AECOPD and admission due to AECOPD. As a secondary purpose of the present study, it was our objective to determine the longitudinal stability of their counts. We analysed the longitudinal stability of the three strata described above from the first to the second visit and from the second to the third visit.

## 2. Materials and Methods

### 2.1. Participants

Participants were recruited between April 2013 and August 2019 from our outpatient clinic, and they were prospectively followed up until May 2020 as part of a hospital-based cohort study [25]. The criteria for inclusion were (1) a diagnosis of stable COPD; (2) age over 50 years; (2) current or former smokers with a smoking history of more than 10 pack-years; (3) chronic fixed airflow limitation defined by fixed ratio, or forced expiratory volume in 1 s (FEV_1_) to forced vital capacity (FVC) of less than 0.7 according to the Global Initiative for Chronic Obstructive Lung Disease (GOLD); (4) regular attendance at the authors’ clinic for more than 6 months to avoid any subsequent changes caused by new medical interventions; and (5) no changes in treatment regimen during the preceding four weeks. Eligible COPD patients had their clinical measures including pulmonary function as well as patient-reported outcomes (PROs) evaluated at entry, and every 6 months thereafter over a 5-year period. When an exacerbation of COPD requiring a change in treatment occurred within 4 weeks of a reassessment day, the evaluation was postponed for at least 8 weeks until the patient recovered. Written informed consent was obtained from all participants.

### 2.2. Measurement

All eligible patients completed the following examinations on the same day. They underwent a routine blood test and pulmonary function tests while sitting including post-bronchodilator spirometry (CHESTAC-8800; Chest, Tokyo, Japan), residual volume (RV) measured by the closed-circuit helium method, and diffusing capacity for carbon monoxide (DL_CO_) measured by the single-breath technique in accordance with guidelines published by the American Thoracic Society and European Respiratory Society Task Force in 2005 [26]. The predicted values for FEV_1_ and vital capacity were calculated according to the proposal from the Japanese Respiratory Society [27]. Participants were also asked to complete the previously validated Japanese versions of the COPD Assessment Test (CAT) [28,29], St. George’s Respiratory Questionnaire (SGRQ) (version 2) [30,31], Hyland Scale and Dyspnoea-12 (D-12) [32,33,34]. They were self-administered under site supervision in the aforementioned order (in a booklet form). Disease-specific health status was assessed using CAT and SGRQ, global health by Hyland Scale and the severity of dyspnoea by D-12.

Outcomes were continuously monitored, and the survival status of all enrolled patients was assessed up until May 2020. The period from entry to the last attendance or death was recorded for the analysis. Acute exacerbation of chronic obstructive pulmonary disease (AECOPD) defined as a worsening of respiratory symptoms that required treatment with oral corticosteroids or antibiotics, or both, and admission due to AECOPD was also recorded throughout the individual follow-up periods. The predictive properties of observational parameters obtained at baseline were analysed in regard to the potential future events of mortality, AECOPD and admission due to AECOPD. To examine the predictive properties, FEV_1_ and the SGRQ Total and CAT scores were also analysed as control indicators [35,36,37,38]. 

On the other hand, to examine the longitudinal stability of BEC, we included all the patients in whom a differential blood cell count was available at all the study visits 1–3 and analysed a sequence of data obtained three times in a row at intervals of 6 to 9 months. The BEC data obtained from participants who missed a visit were excluded from the analysis of longitudinal stability.

### 2.3. Statistical Methods

All results are expressed as mean ± standard deviation (SD) or using median and interquartile range (IQR). A *p* value of less than 0.05 was considered to be statistically significant. Relationships between two sets of data were analysed by Spearman’s rank correlation tests. The significance of between-group differences among non-eosinophilic, intermediate, and eosinophilic groups was determined by Steel–Dwass test and Kruskal–Wallis test. Univariate Cox proportional hazards analyses were performed to investigate the relationships between the clinical measurements at baseline and subsequent events. Results of regression analyses are presented in terms of hazard ratio (HR) with corresponding 95% confidence intervals (CI). Longitudinal stability of BEC was analysed by Friedman test, intraclass correlation coefficient (ICC) and Fleiss’ kappa. ICC values were calculated using both log-transformed and raw data, and interpreted as excellent (≥0.75), good (≥0.60 to <0.75), fair (≥0.40 to <0.60) or poor (<0.40) [39], and Fleiss’ kappa for categorized data as almost perfect (0.81 to 1.00), substantial (0.61 to 0.80), moderate (0.41 to 0.60), fair (0.21 to 0.40) or slight (0.01 to 0.20) [40].

## 3. Results

### 3.1. Cross-Sectional Observation at Baseline

Baseline characteristics of the 135 consecutive patients (123 males) are presented in Table 1. The average age and FEV_1_ were 74.9 ± 6.7 years and 1.70 ± 0.54 L, and 31 patients were current smokers. Eighty-three patients were treated with multiple-inhaler triple therapy, that is, a combination of long-acting muscarinic antagonist (LAMA) and beta2-agonist (LABA) and inhaled corticosteroid (ICS), 33 patients with tiotropium bromide alone, 13 patients with ICS/LABA and 6 patients with no long-acting bronchodilators. While 96 (71.1%) patients were receiving the regimen for treatment including:

ICS at baseline, the BEC was not statistically different between patients taking ICS and those not taking ICS, and some of the other measures were worse in patients with ICS (Appendix A. Table A1).

Spearman’s rank correlation coefficients were obtained to investigate relationships between the BEC at baseline and various factors as shown in Table 1. The BEC was not significantly correlated with clinical, physiological, or patient-reported measures except residual volume (rs = 0.172, *p* = 0.047). In pairwise comparisons of the three groups stratified by BEC (Table 2), there were no significant differences in the measures among non-eosinophilic, intermediate and eosinophilic groups except that for residual volume between the intermediate and eosinophilic groups (*p* = 0.036, Steel–Dwass test). Although the Kruskal–Wallis test was also performed here, there were no significant differences among the three groups.

### 3.2. Predictive Properties of BEC

Of the 135 patients enrolled, 21 patients (15.6%) were confirmed to have died during the follow-up period, which was an average of 41.9 ± 21.8 months, ranging from 3 to 80. The first episodes of AECOPD were identified in 74 out of 130 available patients (56.9%). The duration from entry to the last attendance or the first episode of AECOPD averaged 22.3 ± 19.5 months, ranging from 0 to 79. Thirty-five out of 132 available patients (26.5%) were hospitalized due to AECOPD at least once during the follow-up period of average 31.8 ± 22.6 months, ranging from 2 to 79. Table 3 shows the results from the univariate Cox proportional hazards model in analysing the relationship of the BEC, the three different groups stratified by BEC and the other major clinical measures with mortality, AECOPD and admission due to AECOPD. Almost all the age, FEV_1_, SGRQ Total and CAT Score were significantly strongly related to these types of events but the predictive relationship between age and AECOPD did not reach statistical significance. Neither the BEC nor the three different groups stratified by BEC were significant predictors of subsequent events. A Kaplan–Meier plot for the three different groups stratified by the BEC associated with patient survival is shown in Figure 1. 

### 3.3. Longitudinal Stability of BEC

The mean BEC count was 207 ± 151/mm^3^ at Visit 1 (baseline), 202 ± 125/mm^3^ at Visit 2 and 210 ± 173/mm^3^ at Visit 3 in 86 patients whose counts were available for all three visits (Appendix B. Table A2). There were no significant differences between them (*p* = 0.127, Friedman test). The ICC value was 0.755 (95%CI: 0.647–0.833) using log-transformed data, indicating excellent repeatability while it was 0.596 (95%CI: 0.482–0.698) using raw data, suggesting it was fair. To assess the reliability of agreement between three consecutive measures when assigning to strata, Fleiss’ kappa was calculated to be 0.464, indicating moderate agreement.

At Visit 1 the number of patients in the non-eosinophilic, intermediate, and eosinophilic groups were 20 (23.3%), 48 (55.8%) and 18 (20.9%), respectively (Appendix B. Table A2). The changes between strata over consecutive visits and the resulting distributions are shown in Figure 2. Eleven patients (13%) were persistently non-eosinophilic at all three study visits, but only eight of the patients (9%) were continuously eosinophilic. On the other hand, 26 (30%) patients remained intermediate throughout the period.

## 4. Discussion

In the present study, the predictive properties of BEC were examined for mortality, AECOPD and admission due to AECOPD using univariate Cox proportional hazards analysis. Although it was clearly demonstrated that typical outcome measures such as FEV_1_ as well as health status measure could predict the future event, the BEC was shown to be a poor predictor. Furthermore, since the cross-sectional relationship of BEC with clinical, physiological outcome markers was also interpreted as almost negative, the discriminative properties of BEC have also not been confirmed. These negative results might give contradictory findings that BEC is a poor predictive biomarker for the response to ICS in the prevention of AECOPD. Although the association of relative eosinophilia with exacerbations in clinical trials may be population or circumstance specific, attention should be paid to the universal fact that BEC is generally considered to be a biomarker for COPD.

To our knowledge, eosinophilic predictor properties of mortality have been examined in a few cohort studies though the findings have been equivocal [16,41,42,43]. The prognostic value was reported to be positive in the CHAIN cohort, the BODE cohort [41] and in two studies including the Korean Obstructive Lung Disease (KOLD) cohort [16,42], all showing all-cause mortality was lower in patients with high eosinophil counts compared with those with values <300 cells/mm^3^. Conversely, the findings were negative in the Initiatives BPCO French cohort which was in agreement with our own results (Table 4) [43].

Most cohort studies have continued to pay attention to the association between BEC and AECOPD [21], but they have also provided inconsistent results. While some have reported a positive association between BEC with COPD exacerbation frequency [18,44,45,46], other cohort studies have reported that there was no evidence of such an association (Table 4) [20,41,43,47]. Although most previous studies were designed to statistically compare the frequency of AECOPD between groups, the period from baseline to the first exacerbation is intended to be analysed using the univariate Cox proportional hazards model. Although the present study had the smallest sample size and thus be a potential weak point (Table 4), the following differences among the studies may also have both positive and negative influences on the results; cut-off levels of the BEC, methods of comparison, a definition of the AECOPD and study periods. Thus, it is not easy to compare the results obtained from different studies, and the relationship between BEC and AECOPD has not been established even in the literature. 

On the other hand, some baseline clinical characteristics of COPD according to eosinophil levels have also been reported. Recent meta-analysis revealed that men, ex-smokers, individuals with a history of ischemic heart disease, and individuals with a higher body mass index (BMI) were at higher risk of eosinophilic COPD [48]. Regarding more COPD-specific health outcome measures, the findings of the SPIROMICS cohort showed that at baseline, the high blood eosinophil group had slightly increased airway wall thickness, higher SGRQ Symptom scores, and increased wheezing, but no evidence of an association with the other indices of COPD severity, such as emphysema measured by CT density or the CAT [47]. However, the Initiatives BPCO French cohort group reported that SGRQ Total score was more impaired in lower eosinophilic categories [43]. Korean investigators found that the high group had a longer six-minute walk distance, higher body mass index, lower emphysema index measured by CT and higher inspiratory capacity/total lung capacity ratio (IC/TLC) [42]. In the present study, from cross-sectional observation at baseline, the relationship between baseline characteristics and the BEC was negative and comparison of clinical indices between eosinophilic, intermediate and non-eosinophilic groups classified by BEC was not significantly different except for residual volume, a result that may be related to the Korean findings regarding IC/TLC. 

While it has been reported that BEC is a consistently positive predictor of ICS response, both discriminative and predictive properties were all non-significant’ in the present study. Our result may lead to doubts about the utility of BEC as a COPD biomarker. However, a couple of cohort studies have reported the same findings although there has been some inconsistency. Some recent review articles have suggested a possible reason in that randomized controlled trials focus more on frequent exacerbators than cohort studies including patients with no prior exacerbation history, and that differences in the included participants may produce the inconsistent results.

The longitudinal stability of BEC with a Fleiss’ kappa of 0.464 and an ICC of 0.755 using log-transformed data was considered to be moderate or excellent, showing similar findings to previous studies. While Fleiss’ kappa has not been reported, ICC has been reported to be 0.87–0.89 by Southworth et al. [17], 0.84 by Long et al. [19], 0.57 by Yun et al. [18] and 0.55 by Yoon et al. [23]. Although it is reported that log-transformed data were used to calculate ICC in the former two studies, this was not the case in the latter two manuscripts. The recent GOLD documents state that a threshold BEC of ≥300 /mm^3^ can be used to identify patients with the greatest likelihood of treatment benefit with ICS. In the present study nearly 20% of patients were assigned to this eosinophilic group at every visit, but only 9% were continuously eosinophilic over three visits. Of 18 patients assigned to the eosinophilic group at Visit 1, one was changed to the non-eosinophilic group at Visit 2 and another one at Visit 3. This might suggest that 5–6 % of the eosinophilic group can be subsequently subject to change to the non-eosinophilic group. On the hand, almost a quarter were assigned to the non-eosinophilic group at Visit 1, but 13% remained in the non-eosinophilic throughout all visits. Of 20 patients assigned to the non-eosinophilic group at Visit 1, none were subsequently changed to the eosinophilic group at Visits 2 and 3. 

Greulich T et al. also reported the absolute number and percentage of patients according to three groups defined by different thresholds (150 and 300 cells/mm^3^) at Visits 1, 2 and 3 [24]. They reported that nearly 5% were continuously eosinophilic defined by ≥300 cells/mm^3^ over three visits, but 26% remained non-eosinophilic defined by <150 cells/mm^3^ throughout all visits. It may not be easy to compare the results between studies due to different thresholds. Therefore, in the use of BEC as a biomarker to guide the use of ICS therapy for exacerbation prevention, although the treatment choice should not be based on a one-off BEC measure, the first categorization may often be correct. Of course, the measurement should be repeated even after the determination of the ICS. 

Some limitations of the present study should be mentioned. Most of the issues are related to the study design. First, the present study was limited by the small sample size and distinct male preponderance of the participants. Although the latter is typically observed in patients with COPD in Japan, generalization of these results to women with COPD may be uncertain. This study design might exhibit selection bias because we recruited only patients who could attend our outpatient clinic on a regular basis. It is likely that we did not include enough of those patients without any subjective symptoms who were unaware of having COPD, or patients who could not regularly attend our clinic due to the heavy physical burden. A small proportion of patients with severe or very severe COPD in the present single-centre study might cause a bias. Furthermore, Mathioudakis AG et al. recently conducted a post hoc analysis of ISOLDE and found that the BEC change after ICS administration may predict clinical response to ICS therapy [49]. This hypothesis may need validation in prospectively designed studies but is inconsistent with the present study which found that the BEC was not statistically different between patients taking ICS and those not taking ICS.

## 5. Conclusions

Previous cohort studies evaluating BEC as a mortality or exacerbation predictor have provided inconsistent results. Although most studies were designed to statistically compare the frequency of AECOPD between groups, the period from baseline to death, the first exacerbation and admission due to AECOPD are intended to be analysed using univariate Cox proportional hazards model in a hospital-based cohort study. Almost all the age, FEV_1_, SGRQ Total and CAT Score were significantly strongly related to these types of events, but the predictive relationship between age and AECOPD did not reach statistical significance. Neither BEC nor the three different groups stratified by BEC were significant predictors of subsequent events. As for longitudinal stability, the ICC value was 0.755 using log-transformed data, suggesting excellent, and in the case of assigning with strata, Fleiss’ kappa was calculated to be 0.464, indicating moderate agreement. The predictive properties of BEC may be limited in a real-world Japanese clinical setting. Attention must be paid to the fact that the longitudinal stability of the three strata is regarded as moderate.

## Figures and Tables

**Figure 1 diagnostics-11-00404-f001:**
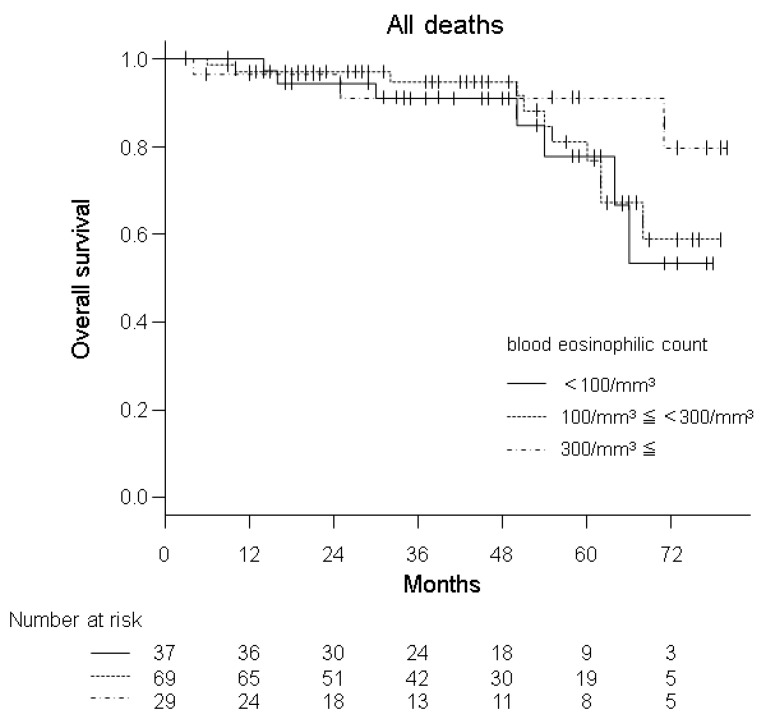
Kaplan–Meier survival curves based on three strata (non-eosinophilic, intermediate and eosinophilic groups) defined by BEC at baseline.

**Figure 2 diagnostics-11-00404-f002:**
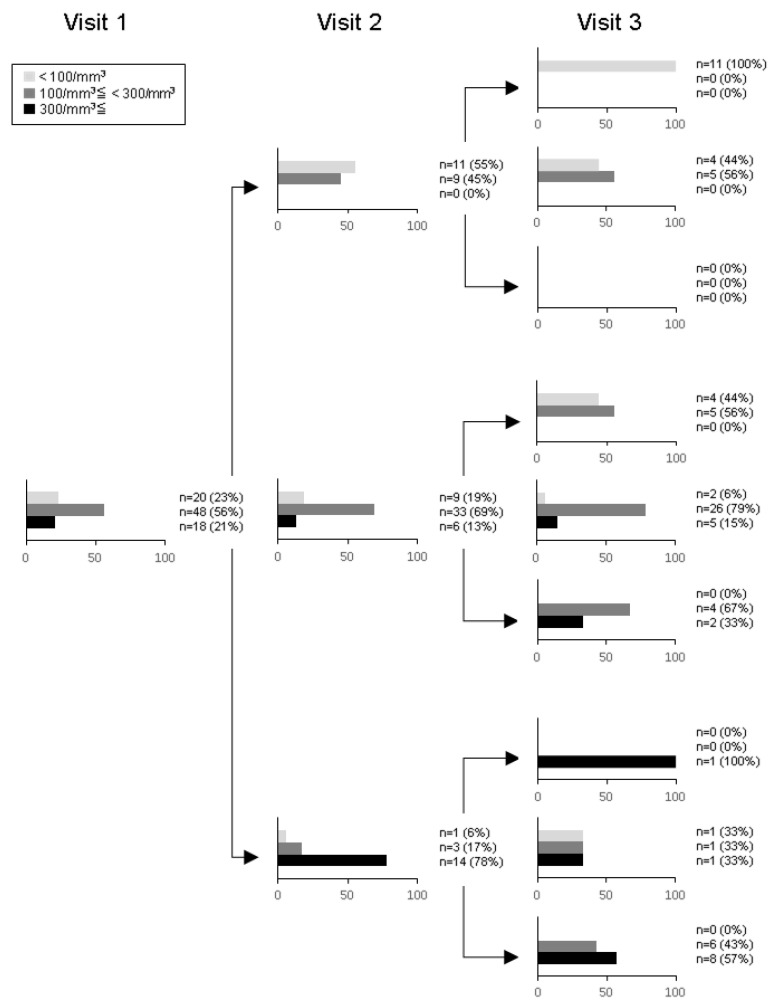
The number and percentage of patients according to three strata (non-eosinophilic, intermediate and eosinophilic groups) defined by BEC at Visits 1, 2 and 3. The proportions of patients at Visit 1 are sequentially subdivided according to their BEC at Visit 2, and the proportions of patients at Visit 2 are sequentially subdivided according to their BEC at Visit 3.

**Table 1 diagnostics-11-00404-t001:** Baseline characteristics in 135 patients with chronic obstructive pulmonary disease (COPD) and Spearman’s rank correlation coefficients with the blood eosinophil count (BEC).

		Median	IQR	Max	Min	Correlations	
						With BEC	
						rs	*p* Value
Age	years	74.0	71.0–80.0	89	51	-	-
BMI	kg/m^2^	22.6	20.3–24.4	35.7	14	-	-
Cumulative Smoking	pack-years	53.0	37.5–73.5	204	10	-	-
FVC	% pred.	97.1	82.2–108.5	141.1	53.7	-	-
FEV_1_	Litres	1.70	1.34–2.09	3.13	0.53	-	-
FEV_1_/FVC	%	58.8	48.4–64.5	69.9	25	-	-
RV ^§^	% pred.	117.1	94.3–139.3	718.9	28.4	0.172	0.047
RV/TLC ^§^	%	44.6	38.3–51.1	85.1	18.1	-	-
DLco ^¶^	% pred.	52.4	39.4–63.2	156.1	10.7	-	-
PaO_2_ ^(1)^	mmHg	78.2	71.4–83.1	101.5	52.1	−0.171	0.047
BNP ^(2)^	pg/mL	25.5	10.8–49.9	229.9	5.7	-	-
SGRQ Total Score	(0–100)	21.0	9.4–35.1	77.3	1.2	-	-
CAT Score	(0–40)	8.0	4.0–14.0	28	0	-	-
Hyland Scale Score	(0–100)	65.0	55.0–75.0	95	20	-	-
D-12 Total Score^§^	(0–36)	1.0	0.0–2.0	24	0	-	-

^(1)^ One patient receiving oxygen, ^(2)^ <5.8 pg/mL considered as 5.7 pg/mL in ten patients, ^§^
*n* = 134, ^¶^
*n* = 133, Missing values of correlation coefficients indicate no statistically significant relationship. IQR, interquartile range; BEC, blood eosinophil count; SGRQ, the St. George’s Respiratory Questionnaire; CAT, the COPD Assessment Test; D-12, Dyspnoea-12. The numbers in parentheses denote possible score range.

**Table 2 diagnostics-11-00404-t002:** Comparison of clinical indices between eosinophilic, intermediate and non-eosinophilic groups classified by BEC.

		Non-Eosinophilic Group		Intermediate Group		Eosinophilic Group	
		*n* = 37		*n* = 69		*n* = 29	
Blood Eosinophil Count (/mm^3^)		<100		≥100 and <300		≥300	
Age	years	74.0	(72.0–80.0)	74.0	(72.0–80.0)	73.0	(69.0–79.0)
BMI	kg/m^2^	22.6	(19.5–24.2)	22.8	(20.8–24.9)	21.8	(20.5–23.8)
Cumulative Smoking	pack-years	54.0	(37.5–78.8)	51.0	(38.0–63.0)	50.0	(40.0–71.8)
FVC	% pred.	100.4	(87.2–108.5)	97.4	(82.2–109.8)	91.8	(78.0–104.3)
FEV_1_	Liters	1.62	(1.38–1.98)	1.71	(1.38–2.10)	1.62	(1.21–2.07)
FEV_1_/FVC	%	59.1	(48.4–66.7)	60.8	(51.6–64.4)	55.8	(43.8–63.4)
RV ^§^	% pred.	123.9	(93.1–137.2)	109.7	(91.7–137.5)	121.9	(115.4–147.7) ***
RV/TLC ^§^	%	44.4	(39.8–49.7)	43.3	(37.6–50.6) ^‡‡^	44.9	(41.0–56.0) ***
DLco ^¶^	% pred.	48.5	(39.3–59.2)	55.3	(41.1–66.8) **	46.7	(33.7–64.3) ***
PaO_2_ ^(1)^	mmHg	79.3	(72.8–87.1)	77.2	(70.7–81.8)	75.8	(70.8–82.1)
BNP ^(2)^	pg/mL	27.8	(10.6–46.4)	25.1	(14.1–49.9)	20.4	(8.4–46.7)
SGRQ Total Score	(0–100)	19.7	(9.4–28.5)	21.5	(8.9–34.8)	25.3	(16.3–40.8)
CAT Score	(0–40)	8.0	(3.0–12.0)	8.0	(4.0–12.0)	9.0	(5.0–15.0)
Hyland Scale Score	(0–100)	65.0	(60.0–75.0)	70.0	(65.0–80.0)	65.0	(50.0–75.0)
D-12 Total Score ^§^	(0–36)	0.5	(0.0–2.0) *	0.0	(0.0–1.0)	1.0	(0.0–2.0)

Data are presented as median (IQR). ^‡‡^
*p* < 0.05 versus eosinophilic group (Steel–Dwass test). No significant difference among three groups with the Kruskal–Wallis test. ^(1)^ One patient receiving oxygen, ^(2)^ < 5.8 pg/mL considered as 5.7 pg/mL in ten patients, ^§^
*n* = 134, ^¶^
*n* = 133, * *n* = 36, ** *n* = 68, *** *n* = 28. IQR, interquartile range; SGRQ, the St. George’s Respiratory Questionnaire; CAT, the COPD Assessment Test; D-12, Dyspnoea-12. The numbers in parentheses denote possible score range.

**Table 3 diagnostics-11-00404-t003:** Univariate Cox proportional hazards analyses on the relationship between major clinical measurements and future events.

			All Deaths (*n* = 135)		AECOPD (*n* = 130)		Admission Due to AECOPD (*n* = 132)	
			Hazard Ratio (95% CI)	*p* Value	Hazard Ratio (95% CI)	*p* Value	Hazard Ratio (95% CI)	*p* Value
Blood eosinophil count		/mm^3^	0.999 (0.995–1.002)	0.352	1.000 (0.999–1.001)	0.915	1.000 (0.998–1.002)	0.869
Three different groups of blood eosinophil count	<100/mm^3^ (Ref.)		1		1		1	
	≥100/mm^3^ and <300/mm^3^		0.849 (0.329–2.192)	0.735	1.285 (0.735–2.247)	0.379	1.289 (0.564–2.946)	0.547
	≥300/mm^3^		0.461 (0.118–1.803)	0.266	1.503 (0.773–2.921)	0.230	1.445 (0.542–3.854)	0.462
Age		years	1.098 (1.025–1.176)	0.007	1.040 (1.000–1.081)	0.050	1.091 (1.027–1.158)	0.005
FEV_1_		Litres	0.293 (0.126–0.679)	0.004	0.318 (0.195–0.519)	<0.001	0.127 (0.061–0.263)	<0.001
SGRQ Total Score		(0–100)	1.023 (1.001–1.047)	0.043	1.028 (1.014–1.043)	<0.001	1.047 (1.027–1.067)	<0.001
CAT Score		(0–40)	1.067 (1.013–1.125)	0.015	1.066 (1.030–1.103)	<0.001	1.144 (1.089–1.203)	<0.001

AECOPD, acute exacerbation of COPD; SGRQ, the St. George’s Respiratory Questionnaire; CAT, the COPD Assessment test.

**Table 4 diagnostics-11-00404-t004:** The predictive properties of blood eosinophil count in subjects with COPD in the literature.

Publication Year	Reference	First Author	The Name of the Cohort or Database	Association with Mortality	Association with AECOPD
2016	#45	Vedel-Krogh S	the Copenhagen General Population Study (*n* = 4.303)	N.A.	positive
2017	#41	Casanova C	the CHAIN cohort (*n* = 424) and BODE cohort (*n* = 308)	positive	negative
2017	#47	Hastie AT	the SPIROMICS cohort (*n* = 2.499)	N.A.	negative
2017	#43	Zysman M	Initiatives BPCO French cohort (*n* = 458)	negative	negative
2018	#16	Shin SH	the Korean Obstructive Lung Disease cohort (*n* = 299)	positive	N.A.
2018	#42	Oh YM	the Korean Obstructive Lung Disease cohort (*n* = 395) and COPD in Dusty Area cohort of Kangwon University Hospital (*n* = 234)	positive	N.A.
2018	#18	Yun JH	the COPDGene (*n* = 1.553) and ECLIPSE (*n* = 1.895) studies	N.A.	positive
2019	#46	Vogelmeier CF	the UK Clinical Practice Research Datalink (*n* = 15,364) and US Optum Clinformatics™ Data Mart databases (*n* = 139,465)	N.A.	positive
2020	#20	Miravitlles M	a primary care electronic medical record database in Catalonia, Spain (*n* = 57,209)	N.A.	negative
2020	#44	Tashiro H	retrospective medical records at the Saga University Hospital (*n* = 481)	N.A.	positive
The present study		Nishimura K	the hospital-based cohort at NCGG, Japan (*n* = 135)	negative	negative

AECOPD, acute exacerbation of COPD; *n*, the number of patients with COPD; N.A., not available.

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
