# Peer review of "Is Blood Eosinophil Count a Biomarker for Chronic Obstructive Pulmonary Disease in a Real-World Clinical Setting? Predictive Property and Longitudinal Stability in Japanese Patients"

_diagnostics, 2021, doi:10.3390/diagnostics11030404_

Round 1

Reviewer 1 Report

The authors revised the manuscript in accordance with the comments of the reviewer, in particular, they switched to nonparametric methods of statistics. I believe that in its current form the article can be recommended for publication.

Reviewer 2 Report

I am happy with the changes and suggest acceptance.

This manuscript is a resubmission of an earlier submission. The following is a list of the peer review reports and author responses from that submission.

Round 1

Reviewer 1 Report

Currently, a fairly large number of studies are devoted to counting the number of eosinophils in COPD. At the same time, the research results boil down to the fact that with an increase in the number of eosinophils over 300, it is accompanied by a greater number of relapses and a better response to corticosteroid therapy. The study group in this case is not homogeneous (123 men out of 135 participants). It is not clear what caused the choice of parametric statistics for processing the results (mean value, standard deviation, confidence interval). To compare 3 groups (Table 2), it was necessary to use the Bonferroni correction or use multivariate methods (for example, the calculation of the Kruskal-Wallis test). Perhaps the small sample size did not allow the authors to identify a statistically significant effect of the number of eosinophils. Why hasn't multivariate regression analysis been performed (Table 3)? Maybe it makes sense to choose a different threshold value for the number of eosinophils? I have seen works where 340 units / μL was used as the threshold. Figure 2 seems interesting to me, but there is little discussion of it in the text. The study group is too small, but it would be interesting to compare the rates of relapse and survival for patients who showed the same level of eosinophils within 3 visits and those in whom it changed. In Table 4, I would add the sample size on which the research was conducted.

Reviewer 2 Report

I have read the article Nishimura et al. with great interest. They evaluated the predictive value of BEC in their longitudinal cohort and found only a very limited one. These findings are not surprising to me as the BEC should be interpreted in the clinical context (i.e. exacerbation burden) and acted upon it.

Comments:

  • Line 46. GOLD is not a guideline. In addition, if you read carefully, eosinophils are advertised in the context of high exacerbation burden (group D and later when escalating depending on exacerbations). Please clarify.
  • You may consider citing https://pubmed.ncbi.nlm.nih.gov/32108044/ which reported that changes due to ICS rather than absolute BEC values predict response to ICS.
  • Please, provide power calculations.
  • Baseline characteristics. It would be important to know the exacerbation history in the preceding 12 months. This is a strong predictor for further events.
  • Was there any difference in the outcomes if the subjects off ICS were investigated separately?
  • The authors should add more data on patients’ medications.
  • Do you have any data on smoking? Did this affect the results?
  • Lines 291-292. “AECOPD, acute exacerbation of COPDSouthworth et al. [17], 0.84 by Long et al. [19], 291 0.57 by Yun et al. [18] and 0.55 by Yoon et al [23].” Please, rephrase.
  • Regarding the association between FEV1 and mortality the authors may consider citing https://pubmed.ncbi.nlm.nih.gov/32547001/ which is the largest study evaluating this relationship.